# The influence of the descending pain modulatory system on infant pain-related brain activity

**Sezgi Goksan[1,2], Luke Baxter[1,2], Fiona Moultrie[1,2], Eugene Duff[1,2], Gareth Hathway[3], Caroline Hartley[1,2], Irene Tracey[2], Rebeccah Slater[1,2]\***

[1]Department of Paediatrics, University of Oxford, Oxford, United Kingdom; [2]Wellcome Centre for Integrative Neuroimaging, FMRIB, Nuffield Department of Clinical Neurosciences, University of Oxford, Oxford, United Kingdom; [3]School of Life Sciences, The University of Nottingham, Nottingham, United Kingdom

**Abstract** The descending pain modulatory system (DPMS) constitutes a network of widely distributed brain regions whose integrated function is essential for effective modulation of sensory input to the central nervous system and behavioural responses to pain. Animal studies demonstrate that young rodents have an immature DPMS, but comparable studies have not been conducted in human infants. In Goksan et al. (2015) we used functional MRI (fMRI) to show that pain-related brain activity in newborn infants is similar to that observed in adults. Here, we investigated whether the functional network connectivity strength across the infant DPMS influences the magnitude of this brain activity. FMRI scans were collected while mild mechanical noxious stimulation was applied to the infant's foot. Greater pre-stimulus functional network connectivity across the DPMS was significantly associated with lower noxious-evoked brain activity (p = 0.0004, r = -0.86, n = 13), suggesting that in newborn infants the DPMS may regulate the magnitude of noxious-evoked brain activity.

DOI: https://doi.org/10.7554/eLife.37125.001

**\*For correspondence:**
rebeccah.slater@paediatrics.ox.ac.uk

**Competing interests:** The authors declare that no competing interests exist.

## Introduction

In adults, pain perception is modulated by the descending pain modulatory system (DPMS), allowing environmental, contextual and cognitive factors to influence our pain experiences (*McMahon et al., 2013*; *Ossipov et al., 2010*; *Tracey and Mantyh, 2007*). The DPMS comprises a network of cortical and subcortical brain regions that can facilitate or inhibit nociceptive afferent brain input via brainstem nuclei (*Ossipov et al., 2010*; *Tracey, 2010*; *Zhuo and Gebhart, 1997*). The functional connectivity of the DPMS is altered in adult chronic pain conditions such as migraine, back pain, fibromyalgia and painful diabetic neuropathy (*Jensen et al., 2012*; *Mainero et al., 2011*; *Segerdahl et al., 2018*; *Yu et al., 2014*), and transient alterations in DPMS connectivity influences pain perception. For example, pre-stimulus functional connectivity between the anterior insula (AI) and the periaqueductal gray (PAG) relates to whether or not a noxious stimulus is perceived as painful (*Ploner et al., 2010*), and pre-stimulus activity in the insular and anterior cingulate cortices (ACC) is predictive of subsequent pain intensity ratings (*Boly et al., 2007*). Furthermore, anticipatory brainstem activity in adults has been shown to predict changes in insula activity evoked by noxious thermal stimulation (*Fairhurst et al., 2007*).

Evidence from animal studies suggests that infant descending pain modulation is immature (*Hathway et al., 2009*). During the first 3 postnatal weeks, anatomical descending projections to the dorsal horn are physically present; however, physiological inhibition of nociceptive input is ineffective or absent in rat pups (*Hathway et al., 2009*; *Fitzgerald and Koltzenburg, 1986*; *Hathway et al.,*

*2006*). Moreover, the brainstem nuclei in the rostral ventromedial medulla (RVM), which are the principle source of these projections, exclusively facilitate nociceptive spinal activity, rather than exerting more adult-like biphasic inhibitory and facilitatory nociceptive control (*Hathway et al., 2009*; *Schwaller et al., 2017*). In the human infant, spinal reflexes are uncoordinated, exaggerated and prolonged (*Andrews and Fitzgerald, 1994*; *Cornelissen et al., 2013*; *Hartley et al., 2016*). Nociceptive reflexes are refined postnatally in infants born prematurely, and by term age, infant reflexes have lower amplitude and shorter duration compared with premature infants (*Cornelissen et al., 2013*; *Hartley et al., 2016*). During this developmental period, the postnatal refinement of spinal cord excitability is concomitant with the maturation of nociceptive brain activity (*Hartley et al., 2016*), leading to the possibility that in the newborn term infant, the brain regions involved in descending pain modulation may be influential in modifying pain behaviour and experience.

In our previous paper, we used fMRI to demonstrate that patterns of noxious-evoked brain activity in the infant are similar to those observed in the adult, and include both sensory and affective brain regions (*Goksan et al., 2015*). Given we cannot measure subjective pain experience in non-verbal infants, we are reliant on objective surrogate measures such as changes in noxious-evoked BOLD activity to make inferences about pain experiences (*Baumgärtner et al., 2010*; *Lee et al., 2008*; *Maihöfner and Handwerker, 2005*). Using this approach, provides the opportunity to investigate whether the network connectivity strength between brain regions involved in descending pain modulation modifies infant pain. The aim of this study was to test the hypothesis that in the human infant, the magnitude of noxious-evoked brain activity recorded using fMRI in response to a standardised nociceptive stimulus is related to the pre-stimulus functional connectivity of brain regions known to comprise the DPMS.

## Results and discussion

### Pre-stimulus functional connectivity in the infant DPMS

Mild experimental noxious stimulation was applied to the infant's foot using a 128 mN PinPrick stimulator. To ascertain the pre-stimulus functional connectivity, we extracted the demeaned BOLD signal from the three volumes recorded immediately prior to the application of the stimulus, which were acquired within the 10 s pre-stimulus period (see Materials and methods and *Figure 3—figure supplement 2A*). We extracted these time courses for the DPMS Network, and for brain regions in two control networks - the first, referred to as the 'Control Network' has similar topography to the DPMS Network, and the second network is a well-recognised resting state network (the Default Mode Network). The DPMS Network comprised the bilateral AI, ACC, amygdala (AMY), RVM, PAG, and the middle frontal gyri (mFG) situated within the dorsolateral prefrontal cortex (*Figure 1A*). This includes the main brain structures identified in the adult DPMS (*McMahon et al., 2013*; *Schweinhardt and Bushnell, 2010*). The Control Network comprised a set of brain regions that are not reported to be involved in descending pain modulation, but included distinct cortical, subcortical and brainstem structures, and had similar topographic distribution to the DPMS brain regions. Whilst the Control Network is not a known functional network within the brain, this network controls for global signal confounds for example respiratory or cardiovascular signals. The brain regions in the Control Network are the bilateral calcarine cortices (CAL), caudate (CAU), hippocampus (HIP), pontine nuclei (PON), recti gyri (RGY) and the supplementary motor areas (SMA) (*Figure 1C*). As an additional control, the Default Mode Network (an established network that has been identified in adults and term infants) (*Doria et al., 2010*; *Raichle, 2015*) allowed us to test the specificity of the relationship between the pre-stimulus functional connectivity of the DPMS and the noxious-evoked BOLD activity. The Default Mode Network included the posterior cingulate cortex (PCC), the inferior parietal lobules (IPL) and the medial superior frontal gyrus (mSFG) situated within the medial prefrontal cortex (mPFC) (*Figure 1E*).

The overall mean pre-stimulus functional connectivity was calculated for each network (*Figure 1B, D,F*). This was not significantly different between the DPMS Network and the Control Network (mean pre-stimulus functional connectivity: DPMS Network = 0.08 ± 0.10; Control Network = 0.15 ± 0.12). Unsurprisingly, given that the Default Mode Network is a canonical network

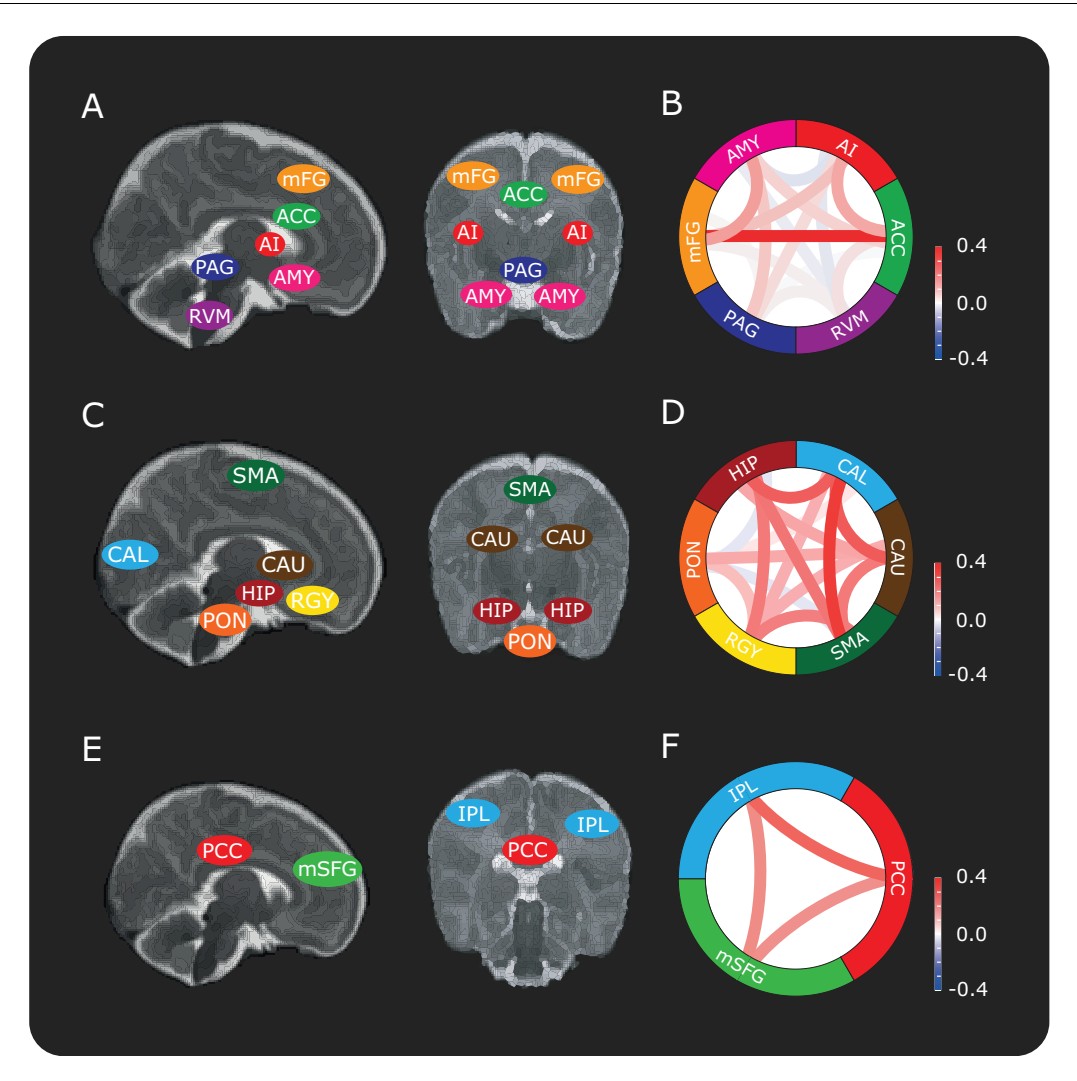

**Figure 1.** Connectivity between brain regions in the DPMS and control networks. Schematic representation showing approximate locations of brain regions in sagittal and coronal slices in the (**A**) DPMS Network, (**C**) Control Network and (**E**) Default Mode Network. Each anatomical region of interest is identified in *Figure 1—figure supplement 1* and the source data is provided in *Figure 1—source data 1*. *Figure 1—figure supplement 2* shows the registration of two example masks from template to functional space and example time series. Network schematics of the mean pre-stimulus functional connectivity between pairs of regions in the (**B**) DPMS Network, (**D**) Control Network and (**F**) Default Mode Network. For abbreviations see main text.

DOI: https://doi.org/10.7554/eLife.37125.002

The following source data and figure supplements are available for figure 1:

**Source data 1.** All region-of-interest masks in standard space.
DOI: https://doi.org/10.7554/eLife.37125.006
**Figure supplement 1.** Masks of regions included in the DPMS, Control Network and Default Mode Network.
DOI: https://doi.org/10.7554/eLife.37125.003
**Figure supplement 2.** Registration and time series data.
DOI: https://doi.org/10.7554/eLife.37125.004
**Figure supplement 3.** Pre-stimulus connectivity is stable.
DOI: https://doi.org/10.7554/eLife.37125.005

that has been identified in both adult and infant resting state data (*Doria et al., 2010*; *Raichle, 2015*), the functional connectivity of the Default Mode Network was significantly greater (mean pre-stimulus functional connectivity: Default Mode Network = 0.24 ± 0.18) than connectivity within the DPMS and the Control Network (p = 0.0014, repeated measures ANOVA, Tukey post-hoc

comparison of DPMS and Control Network: p = 0.06, Default Mode Network and DPMS: p < 0.001, Default Mode Network and Control Network: p = 0.047).

## Characterisation of noxious-evoked brain activity in infants

Consistent with previous reports (*Goksan et al., 2015*; *Williams et al., 2015*), we identified positive clusters of noxious-evoked BOLD activity in the bilateral postcentral gyrus (somatosensory cortices), thalamus, anterior cingulate cortex and contralateral posterior insular cortex (*Figure 2*, *Table 1*). We report a reduction in the number of active brain regions compared with our previous publication (*Goksan et al., 2015*), and demonstrate more highly localised clusters of significant activity within distinct anatomical regions (*Figure 2*). For example, clusters of activity can now be identified in the medial surface of the somatosensory cortex, which encodes the somatotopic foot representation (*Figure 2* and *Figure 2—source data 1*). These differences have arisen due to improvements in the data analysis pipeline to incorporate recent recommendations and methodological advances. Importantly, the statistical cluster-defining threshold has increased from z = 2.3 to z = 3.1, to account for potential inflation in family wise error rates that have been observed across a broad range of MRI studies (*Eklund et al., 2016*). Improved filtering of head motion parameters using FIX (*Griffanti et al., 2014*; *Salimi-Khorshidi et al., 2014*) and an infant-specific haemodynamic response function (*Arichi et al., 2012*) were also used (see Materials and methods). The brain regions identified in this more stringent analysis represent the most robustly activated clusters of noxious-evoked brain activity in the infant, and are consistent with those most commonly reported in adults (*Tracey and Mantyh, 2007*). Our previous report that the infant pattern of pain-related brain activity is similar to that observed in adults is reconfirmed here (*Goksan et al., 2015*).

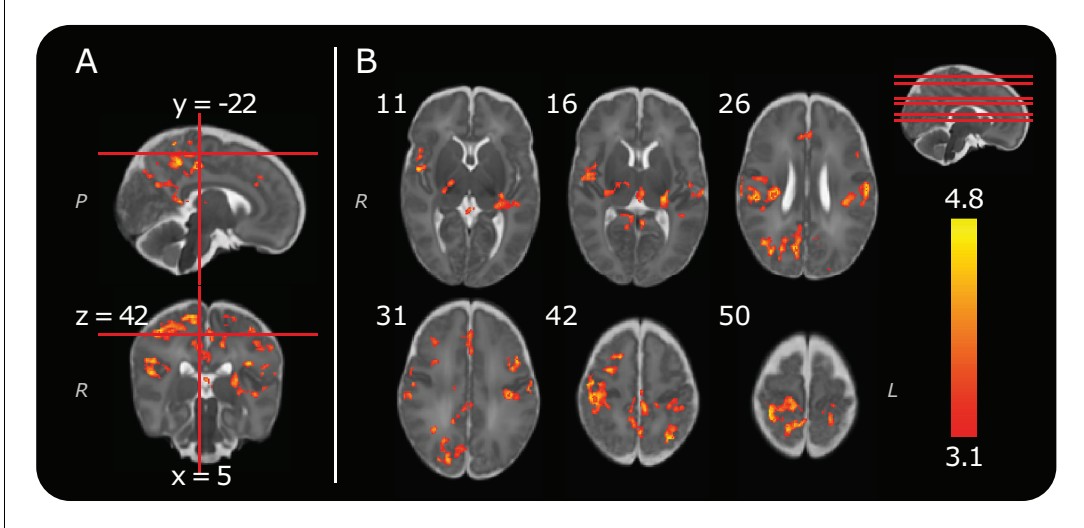

**Figure 2.** Group noxious-evoked brain activity. (**A**) Sagittal and coronal views of the significant group activity from the 13 infants. Red lines indicate how the two images (and the transverse image at z = 42, in B) relate to one another. (**B**) Transverse images showing significant group activity. The source data is provided in *Figure 2—source data 1*). Numbers by the top left of each image represent coordinate locations in infant template space. The location of each transverse slice is demonstrated (red lines) on the sagittal template brain in the top right. The activity map is overlaid on a standard template of an infant brain at 40 weeks' gestational age (*Serag et al., 2012*). Letters in italics depict axis labels: *L* = left, *R* = right, *P* = posterior. Statistical maps are of cluster thresholded z-statistics (z > 3.1, cluster significance threshold p < 0.05).
DOI: https://doi.org/10.7554/eLife.37125.007

The following source data is available for figure 2:

**Source data 1.** Thresholded group activity map.
DOI: https://doi.org/10.7554/eLife.37125.008

**Table 1.** Significant positive clusters of noxious-evoked brain activity, observed across the whole group (n = 13, cluster forming threshold: z = 3.1, cluster significance threshold, p = 0.05).
This table provides an anatomical description and the location of the peak z-statistic within each active brain region. The group activity reported consisted of 14 distinct clusters, some of which spanned multiple brain regions.

| Anatomical description of location of activity | | Maximum z-statistic within cluster | Coordinates of maximum z-statistic in infant template space | | |
| --- | --- | --- | --- | --- | --- |
| | | | X | Y | Z |
| Post-central gyrus | Contra | 4.8 | 7.7 | −27.5 | 50.7 |
| | Ipsi | 4.7 | −23.2 | −27.5 | 43.8 |
| Posterior cingulate sulcus | Contra | 4.8 | 6.9 | −23.2 | 36.0 |
| | Ipsi | 3.8 | −6.0 | −20.6 | 34.3 |
| Superior parietal lobule | Contra | 4.8 | 12.9 | −44.6 | 47.2 |
| | Ipsi | 4.7 | −14.6 | −43.8 | 43.8 |
| Thalamus | Contra | 4.7 | 16.3 | −19.7 | 12.0 |
| | Ipsi | 4.6 | −13.8 | −24.0 | 16.3 |
| Supra-marginal gyrus | Ipsi | 4.7 | −25.8 | −29.2 | 37.8 |
| | Contra | 4.6 | 30.1 | −24.9 | 28.3 |
| Superior frontal sulcus | Contra | 4.7 | 16.3 | −4.3 | 44.6 |
| Middle frontal gyrus | Ipsi | 4.7 | −27.5 | 0.9 | 30.0 |
| | Contra | 4.0 | 24.1 | 5.2 | 33.5 |
| Cuneus | Contra | 4.7 | 7.7 | −53.2 | 26.6 |
| | Ipsi | 4.2 | −5.1 | −50.7 | 23.2 |
| Superior parietal lobe / Precuneus | Contra | 4.7 | 5.2 | −35.2 | 39.5 |
| | Ipsi | 4.0 | −0.9 | −43.8 | 41.2 |
| Pre-central gyrus / Central sulcus | Contra | 4.6 | 24.1 | −14.6 | 45.5 |
| | Ipsi | 3.9 | −8.6 | −23.2 | 53.2 |
| Posterior insula | Contra | 4.6 | 19.0 | −19.7 | 24.0 |
| Parietal operculum | Ipsi | 4.6 | −34.4 | −21.4 | 18.0 |
| Superior temporal gyrus / Posterior operculum | Contra | 4.6 | 31.0 | −27.5 | 22.3 |
| Occipital gyrus | Ipsi | 4.6 | −12.0 | −63.5 | 21.4 |
| Anterior cingulate cortex | Ipsi | 4.2 | −1.7 | 14.7 | 24.0 |
| | Contra | 3.9 | 4.3 | 12.1 | 25.7 |
| Superior temporal gyrus | Ipsi | 4.2 | −27.5 | −28.3 | 12.8 |

DOI: https://doi.org/10.7554/eLife.37125.009

## Relationship between the pre-stimulus functional connectivity of the DPMS and noxious-evoked brain activity

For each infant, the mean pre-stimulus functional connectivity across the DPMS Network and the control networks were calculated, and related to the mean percentage change in BOLD activity evoked by the noxious stimulation (calculated for each individual participant across all the voxels where significant group activity was identified). There was a significant inverse relationship between the magnitude of pre-stimulus DPMS functional connectivity and the percentage change in noxious-evoked BOLD activity (Pearson correlation coefficient (r) = -0.86, p = 0.0004, parameter estimate (β) = -0.74, linear model also included gestational age in weeks as an explanatory variable, *Figure 3A*). Infants with greater functional connectivity across their DPMS Network prior to noxious stimulation had lower noxious-evoked brain activity. In contrast, the mean functional connectivity in the Control Network and in the Default Mode Network were not related to the mean change in noxious-evoked BOLD activity (Control Network: r = -0.36, p = 0.26, β = -0.25; Default Mode Network:

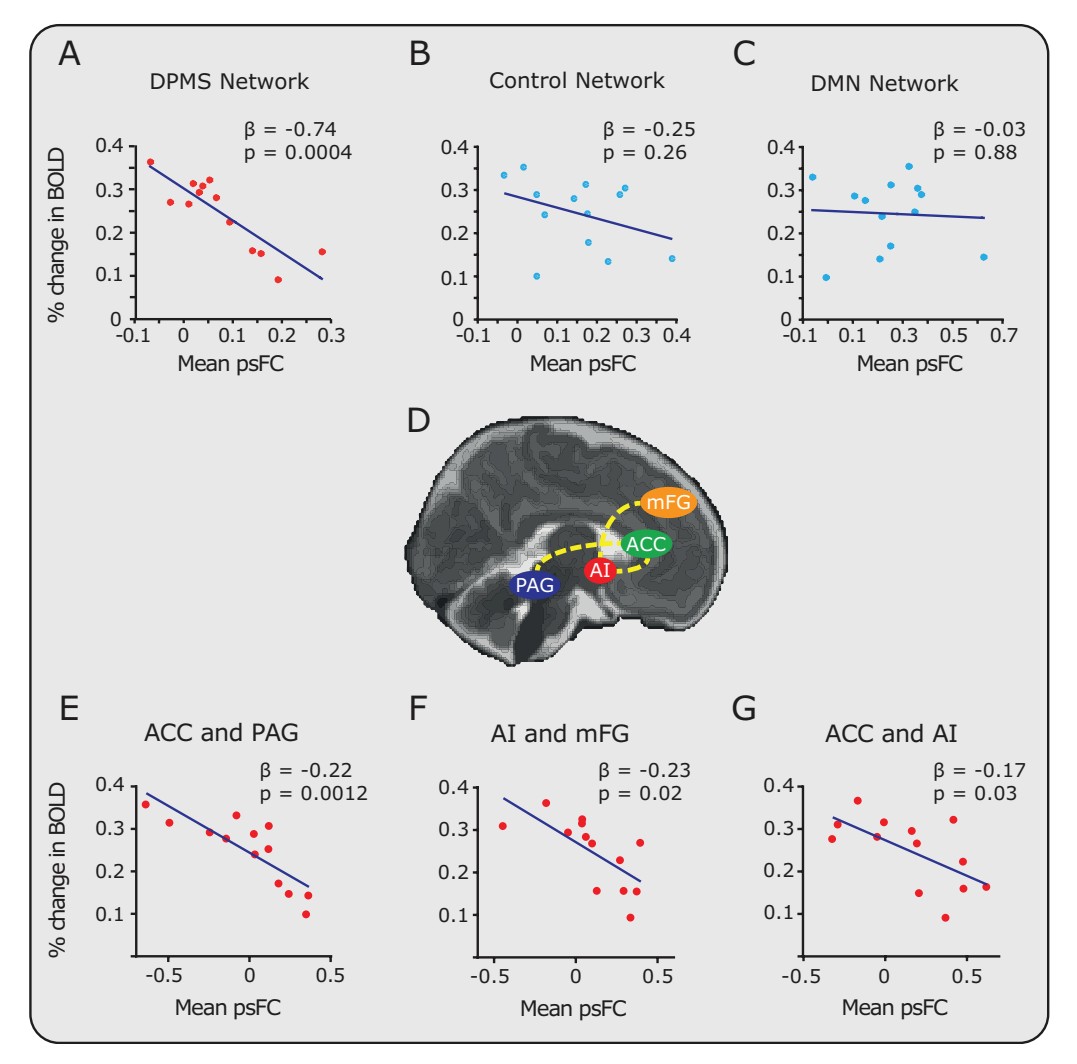

**Figure 3.** Relationship between noxious-evoked brain activity and pre-stimulus functional connectivity in the DPMS and control networks. Linear regression models (blue lines) were used to compare pre-stimulus functional connectivity (psFC) with the percentage change in BOLD activity in the (**A**) DPMS Network, (**B**) Control Network and (**C**) the Default Mode Network (DMN). Noxious-evoked brain activity for each infant (calculated within a mask of the group activity, see *Figure 2*) was adjusted for gestational age (in weeks) at the time of study. Coloured circles represent data from individual infants within the DPMS (red) and control networks (light blue). *Figure 3—source data 1* provides the individual PAG and RVM functional masks for each infant. *Figure 3—figure supplement 1* shows the relationship between the percentage change in BOLD activity and the psFC in the DPMS Network and Control Network with the brainstem regions removed. (**D**) The brain schematic highlights the pairs of brain regions where psFC was significantly correlated with percentage change in the BOLD response (dashed yellow lines). (**E,F,G**) The three pairs of regions within the DPMS Network which demonstrated strong correlations between mean psFC and noxious-evoked brain activity.

DOI: https://doi.org/10.7554/eLife.37125.010

The following source data and figure supplements are available for figure 3:

**Source data 1.** Individual DPMS brainstem masks in functional space.

DOI: https://doi.org/10.7554/eLife.37125.013

**Figure supplement 1.** Relationship between percentage change in noxious-evoked brain activity and pre-stimulus functional connectivity in the DPMS Network and Control Network with the brainstem regions removed.

DOI: https://doi.org/10.7554/eLife.37125.011

**Figure supplement 2.** Example data from individual infants.

DOI: https://doi.org/10.7554/eLife.37125.012

r = 0.06, p = 0.88, β= -0.03 *Figure 3B,C*). The absence of a significant relationship between the functional connectivity of the Default Mode Network and the noxious-evoked BOLD activity suggests that the influence of the DPMS on the noxious-activity is not generalisable across all established brain networks.

To explore the relative contribution of different brain regions within the DPMS Network, the relationship between the functional connectivity and the mean change in noxious-evoked BOLD activity was calculated for each pair of brain regions. Increased pre-stimulus functional connectivity between the ACC and PAG was associated with a substantial reduction in noxious-evoked BOLD activity (adjusted for age, p = 0.0012, β = -0.22, *Figure 3E*). Functional connectivity between the AI-mFG and ACC-AI were also strongly related to the change in noxious-evoked BOLD activity (p = 0.02, β = -0.23 and p = 0.03, β = -0.17 respectively, *Figure 3F,G*). For all other pairs of brain regions, the functional connectivity strength did not influence the magnitude of noxious-evoked brain activity. The observation that a high degree of functional connectivity between the ACC and PAG is strongly associated with a reduction in pain-related brain activity in the infant is interesting in light of observations in adults where greater co-variation in the functional activity of the rostral ACC and PAG relates to an increase in the efficacy of endogenous analgesia elicited by placebo treatment (*Petrovic et al., 2002*). Anticipation of placebo has been associated with greater pre-stimulus activity in the PAG, and leads to a placebo-induced reduction in evoked brain activity in the thalamus and rostral ACC (*Wager et al., 2004*). The importance of the PAG, as part of the DPMS, has also been demonstrated in animal studies, where direct stimulation of the PAG is associated with a reduction in incoming nociceptive information from the peripheral nervous system (*Reynolds, 1969*). In adult rodents, descending modulation (evidenced by PAG activation) preferentially modulates C-fibre input (*McMullan and Lumb, 2006*; *Waters and Lumb, 2008*), whereas the noxious stimulus applied in this study likely preferentially activates A-delta fibres, which may be differentially modulated compared with C fibre input. However, it is not known how other supraspinal components of the DPMS respond to activity in subclasses of nociceptors in humans. Further work is needed to understand the developmental trajectory of the PAG-RVM axis in humans and the maturation of its connections to the spinal cord.

Human and non-human infants display heightened sensitivity to noxious stimulation, which has long been attributed to hyper-excitable spinal reflex networks (*Fitzgerald, 2005*). In the neonatal rodent, brain structures within the DPMS facilitate, rather than inhibit, spinally-mediated nocifensive behaviours (*Hathway et al., 2009*). During human preterm development, these exaggerated reflexes are refined with shorter durations, lower magnitudes, and higher response thresholds, and patterns of noxious-evoked brain activity concomitantly mature (*Hartley et al., 2016*; *Cornelissen et al., 2013*; *Fabrizi et al., 2011*). The data presented here in term infants suggest that the functional DPMS brain networks have an inhibitory function at the level of the brain, similar to that observed in the adult (*Ossipov et al., 2010*). There is, however, a potential contradiction with observations in neonatal rodents where facilitation at the level of the spinal cord has been observed in electrophysiological recordings. This could reflect maturational differences in the connectivity of supraspinal DPMS regions and the connectivity of the descending pathways from the RVM to the spinal nociceptive dorsal horn network in newborn infants. Nevertheless, this interpretation relies on a comparison across species, which is based on a theoretical age-equivalence between human infants and neonatal rat pups. Some networks will likely have a different developmental trajectory in rodents compared with humans and the maturation of the CNS is not a coordinated linear process as different networks likely mature at different rates (*Clancy et al., 2001*). As we have not measured activity in the spinal cord, we cannot determine the relationship between functional connectivity in the DPMS and spinal activity in the term-aged infant.

Collecting functional imaging data in the brainstem is challenging, both in adults and infants, due to head motion, and cardiac-related and respiratory-related motion (*Harita and Stroman, 2017*). In this study, we identified and regressed out physiological noise using independent component analysis (*Salimi-Khorshidi et al., 2014*; *McKeown et al., 2005*; *McKeown et al., 1998*). As further confirmation of the results, we re-assessed the strength of pre-stimulus functional connectivity in the DPMS Network and Control Network excluding brainstem regions; namely the PAG and RVM for the DPMS Network and the PON from the Control Network. The strength of the DPMS Network within the remaining cortical and subcortical regions, the ACC, AI, mFG and AMY, was still significantly inversely related to noxious-evoked brain activity (r = -0.61, p = 0.04, β = -0.37, *Figure 3—*

*figure supplement 1*). As before, the Control Network excluding the PON, was not significantly correlated with noxious-evoked brain activity (r = -0.29, p = 0.36, β = -0.15). This suggests that our results are unlikely to be driven by noise within the brainstem. While there are inherent limitations in this study in terms of the spatial resolution that can be achieved when imaging small structures within the brainstem, we believe that the PAG and RVM masks that we individually defined for each infant are well localised within these anatomical structures. *Figure 3—source data 1* gives the individual PAG and RVM functional masks for each infant.

It is possible that application of the noxious stimulus could influence the pre-stimulus data; however, the stimulus presentation was not predictable, and the time-period between stimuli was always greater than 25 s. The pre-stimulus functional connectivity of the DPMS was not dependent on stimulus number (p = 0.33, repeated measures ANOVA, see *Figure 1—figure supplement 3*), suggesting that the functional connectivity of this network is relatively stable. In adults, functional brain networks are also thought to be dominated by stable individual features, and only modestly influenced by evoked factors and day-to-day variability (*Gratton et al., 2018*). To further understand the relationship between infant noxious-evoked brain activity and the DPMS, functional connectivity analysis of resting state data, and underlying structural connectivity and white matter fibre integrity between DPMS regions using diffusion MRI is warranted (*Gratton et al., 2018*; *Friston, 2011*). Neuroimaging studies in both humans and animals suggest that functional connectivity measures can be used to better understand how networks of brain regions are involved in complex functions (*Cole et al., 2016*; *Fox et al., 2005*; *Smith et al., 2013*), including pain (*Baliki et al., 2014*). While these measures may represent direct or indirect communication between these brain regions (*Fox et al., 2005*; *Smith et al., 2013*), they may also reflect underlying changes in the amplitude of the neural signals, which are unrelated to neural synchrony (*Friston, 2011*; *Cole et al., 2016*; *Duff et al., 2018*) – further investigation of DPMS structural and functional connectivity may elucidate these underlying mechanisms.

In summary, this study suggests that in term infants the DPMS may be influential in regulating the magnitude of noxious-evoked brain activity. In adults, greater pre-stimulus activity in brain regions within the DPMS network are coupled with lower behavioural pain reports (*Ploner et al., 2010*; *Boly et al., 2007*; *Fairhurst et al., 2007*). Therefore, a possible interpretation of our results is that when regions within the DPMS are more strongly functionally connected, infants have a greater ability to regulate their pain experience and dampen the magnitude of their brain activity in response to incoming nociceptive input. Surgical injury in the neonatal period is known to lead to whole-body changes in pain sensitivity that persist into childhood (*Walker et al., 2009*), and this may be dependent upon changes in the maturation of the DPMS, especially the RVM (*Walker et al., 2015*). To understand how the DPMS develops during early life, and how it is influenced by early life experiences, further investigation of the DPMS is required in both younger preterm infants and older infants. For example, it has been suggested that development of aberrant DPMS function in early life may lead to long-term vulnerability towards chronic pain states (*Denk et al., 2014*). The presence of a functional supraspinal modulation system in a term-aged human infant is consistent with the proposal that the emergence of top-down inhibitory pathways develop in early life (*Hartley et al., 2016*). We conclude that the DPMS network can influence the magnitude of pain-related brain activity in term-aged infants.

## Materials and methods

### Participants

Seventeen newborn term-aged infants were recruited from the Maternity Unit at the John Radcliffe Hospital, Oxford, UK. All infants completed the full study protocol. The National Research Ethics Service provided ethical approval: REC reference 12/SC/0447. Informed written parental consent was obtained prior to each study. The study was carried out in accordance with the standards set by the Declaration of Helsinki and Good Clinical Practice guidelines.

Data from four infants were excluded from the analysis because the most caudal region of interest, the rostral ventral medulla in the brainstem, fell outside of the field of view. Therefore, 13 term infants (average gestational age (GA) at study = 40 weeks, range 38 to 43 weeks) were included in this analysis. The average postnatal age at the time of the study was 4 days (range 1 to 8 days). Eight

of the 13 infants included in this analysis were also included in our previous publication (*Goksan et al., 2015*).

## Study protocol

Infant recruitment criteria, experimental study design and MRI study protocol were identical to that described previously by *Goksan et al., 2015*. In brief, all infants were scanned at the Centre for Functional Magnetic Resonance Imaging of the Brain (FMRIB), John Radcliffe Hospital, Oxford. Prior to scanning infants were fed and swaddled and provided with three levels of ear protection: ear putty (Mack's Kids size earplugs, McKeon Products Inc., MI), ear muffs (Minimuffs, Natus Medical Inc., Galway, Ireland) and hearing defenders (Em's 4 Bubs Baby Earmuffs, Em's 4 Kids, Brisbane, Australia), with noise reduction ratings of 22 dB, 7 dB, and 22 dB, respectively. Infants were then placed in a vacuum-positioning mattress and all scanning was done when infants were settled or asleep.

During all MRI sessions, T2-weighted structural images were collected prior to acquisition of functional echo planar imaging (EPI) scans. During individual infant's functional scans, acute experimental noxious stimulation was applied using a calibrated nociceptive stimulator (force: 128 mN, PinPrick Stimulators, MRC Systems). Noxious stimulation was applied 10 times to the heel of the left foot by the same experimenter and with a minimum inter-stimulus interval of 25 s. The interval was chosen based on the neonatal term infant haemodynamic response function (HRF) described by *Arichi et al. (2012)* and the interval was extended if necessary to ensure the infant was settled at the time of stimulation.

## MRI acquisition

Images were collected using a Siemens 3-Tesla Magnetom Verio scanner (Erlangen, Germany) with a 32-channel adult head coil. T2-weighted turbo spin echo structural scans were acquired for each infant (sequence parameters: repetition time/echo time (TR/TE) = 14740/88 ms; flip angle 150 °; resolution 1 mm$^3$; slices = 85, field of view (FOV) = 192×192 mm, acceleration = GRAPPA 2, slice order = interleaved, with no slice overlap). BOLD images were acquired using a T2*-weighted EPI acquisition (sequence parameters: TR/TE = 2500/40 ms; flip angle = 90°; FOV = 192×192 mm; imaging matrix 64×64; resolution 3×3×3 mm; slices = 33, collected in descending order; average total volumes = 142). Prospective Acquisition Correction for head motion (PACE) was applied during all EPI scans (*Thesen et al., 2000*), as described previously in *Goksan et al., 2015*. Field map images were obtained for post-acquisition correction of gradient field effects (sequence parameters: TR = 400 ms; TE1/TE2 = 5.19/7.65 ms; flip angle = 60°; FOV = 192×192 mm; imaging matrix 64×64; resolution 3×3 ×3 mm; slices = 36, slice order = interleaved; inter-slice gap = 0.75 mm). The noxious stimuli were time-locked to the fMRI recording using Neurobehavioural Systems (Presentation, www.neurobs.com) software; coded to detect an experimenter's button-press each time an experimental stimulus was applied to the participant's foot.

## Data analysis
### MR data processing

All MR data pre-processing were done using FMRIB Software Library (FSL) (www.fmrib.ox.ac.uk/fsl), Versions 5.0.10 and 4.1.9. Version 5.0.10 was used to prepare the structural and field map images. FSL's Brain Extraction Tool (BET) was used in order to extract brain-tissue signal from the non-brain structures in each infant's structural image (*Smith, 2002*). The fractional intensity threshold and threshold gradient parameters within BET were adjusted in order to obtain the most accurate brain extraction per subject. A mask of each infant's brain-extracted structural scan was registered to the fieldmap and used to guide fieldmap preparation. All fMRI data registrations were done using FMRI Expert Analysis Tool (FEAT) Version 5.98 (FSL Version 4.1.9) to avoid boundary-based registration (BBR), due to hard coding of the adult-appropriate BBR-slope parameter, which is unsuitable for infant fMRI data. Functional images were registered to a standard average infant template (40 week GA template; downloaded from www.brain-development.org). Each EPI was initially registered to the infant's structural image (FLIRT: rigid body transformation with six DOF [*Jenkinson et al., 2002*; *Jenkinson and Smith, 2001*]). Subsequently, images in structural space were non-linearly registered to the neonatal-specific template image, which corresponded to the GA of the infant at the time of

the study (*Serag et al., 2012*) and then to the standard infant 40-week gestation template (FNIRT: non-linear transformation with twelve DOF).

FEAT (Version 5.98) was used to run functional data pre-processing steps implemented within FSL; which included motion correction of the functional data using MCFLIRT (*Jenkinson et al., 2002*), distortion correction using FUGUE, brain extraction using BET, high pass temporal filtering at 0.01 Hz (100 s period), and grand mean scaling. Spatial smoothing is a common preprocessing step that typically filters the data with a smoothing kernel extent (measured in full width at half maximum) larger than one voxel. Given our spatial resolution, the voxel size relative to the neonatal brain, and our use of small brainstem ROIs, spatial smoothing was deemed inappropriate and thus omitted. MELODIC (model-free fMRI analysis using probabilistic independent component analysis) was used to decompose functional data into spatially independent components, which were subsequently manually labelled as signal or noise (*Griffanti et al., 2017*). FIX (FMRIB's ICA-based Xnoiseifier, v1.065) was then applied to regress out the noise component time series and 24 head motion parameter time series (*Griffanti et al., 2014*; *Salimi-Khorshidi et al., 2014*). While spatial ICA-based denoising does not remove global signal artefacts, we did not include a pre-processing step to address this, such as GSR (global signal regression). We addressed the potential issue of global signal cofounds in the main analysis by including our Control Network.

MR data statistical analysis was conducted using FSL (Version 5.0.10). Time-series statistics were generated using general linear modelling (GLM) in FEAT (Version 6.00). The experimental model was created using an event-related design, where each input represented the timing of each noxious stimulus (duration: approximately 1 s), recorded during the scanning session via the Presentation code. The experimental design was convolved with three term-infant-specific optimal basis functions generated by Arichi and colleagues (*Arichi et al., 2012*). Motion outlier variables were included in the model by identifying volumes where large deviations in head position occurred. Each motion variable was generated by FSL's motion outliers command using the DVARS option; which calculated the rate of change of the variance between volumes (*Power et al., 2012*). Cluster thresholding (with a cluster defining threshold of p = 0.001 (z = 3.1) and a cluster significance threshold of p = 0.05) was used to identify significant increases in BOLD following experimental noxious stimulation.

Group analysis was run in FSL (Version 5.0.10), using mixed effects FLAME 1 and 2 in FEAT (Version 6.00), with automatic outlier detection. The first contrast of the parameter estimate (COPE) statistical image of each participant was input into the higher group analysis, therefore only taking into account the first basis function described by *Arichi et al. (2012)*. This function closely resembles a double gamma function with a peak at 7 s and an undershoot to positive peak ratio of 0.49. Two neonatal-specific atlases, the University of North Carolina's (UNC) atlas (*Shi et al., 2011*) and an Imperial College London (ICL) atlas (*Serag et al., 2012*), and an adult atlas (*Mai et al., 2008*) were used to guide description of the resulting group activity (in *Table 1*). Three atlases were required because each atlas provided varying levels of anatomical specificity. The ICL atlas was the most general, describing all the lobes of the brain, as well as some deep brain nuclei and maps of the CSF, grey and white matter. Despite this broad labelling, the ICL atlas also provided the most accurate partition between anatomical boundaries. The UNC infant atlas was used as it contained more specific anatomical masks. However, the partitions between anatomical boundaries were less good; therefore the ICL atlas was used in conjunction to define boundaries. Finally when describing the anatomical location of peaks within the group activity (see *Table 1*), an adult atlas was used as it provided further guidance for labelling specific gyri and sulci. Clusters that extended across more than one brain region were described separately only when a region of activity with a separate local peak voxel was observed within the adjacent brain region. For all regions named in the table, masks were hand drawn and aimed to include all the active voxels within each region, however given the subjective nature of this task it is possible that small regions of activity that formed part of the same cluster may have been overlooked. The function *Cluster*, available within FSL, was then used to obtain maximum z-statistics and their coordinate locations.

## Pre-stimulus functional connectivity (psFC)

Mean time series were calculated in 15 brain regions. Six regions were identified as key regions within the DPMS: anterior cingulate cortex (ACC), amygdala (AMY), anterior insula (AI), middle frontal gyrus (mFG) (a region within the dorsolateral prefrontal cortex - dlPFC); assessed using the following papers (*Rajkowska and Goldman-Rakic, 1995*; *Sallet et al., 2013*; *Stagg et al., 2013*),

periaqueductal grey (PAG) and rostal ventral medulla (RVM). A further nine brain regions, without a known role in the DPMS, were included within the two control networks. The Control Network included the calcarine cortex (CAL), caudate (CAU), hippocampus (HIP), pons (PON), recti gyri (RGY) and the supplementary motor area (SMA). Three regions were identified in the Default Mode Network – the posterior cingulate cortex (PCC), inferior parietal lobules (IPL) and the medial superior frontal gyrus (mSFG). The mSFG was chosen in place of the medial prefrontal cortex (commonly reported as part of the DMN) because a medial prefrontal cortex mask was not available as part of the UNC or ICL infant atlases; therefore, the mSFG was taken as the representative of this region.

Brain regions commonly reported to be involved in descending pain modulation (*Schweinhardt and Bushnell, 2010*), were included in the DPMS Network. However, this network did not include all DPMS regions, as for example, the hypothalamus also plays a key role in descending pain modulation (*Denk et al., 2014*; *Dafny et al., 1996*). The DPMS Network reported here therefore includes core regions within the adult DPMS network that could be confidently identified and masked in the infant.

Using the two neonatal atlases described above (UNC and ICL), region of interest (ROI) masks were created. ACC, AMY, CAU, HIP and PCC masks were taken directly from the ICL atlas at 40 week GA. CAL, IPL, mFG, mSFG, RGY and SMA were regions within the UNC infant atlas. Following registration of the UNC atlas to the 40-week template brain, masks of all six aforementioned UNC atlas regions were isolated. Subsequently, UNC atlas masks were carefully inspected to ensure that each mask fell within the appropriate boundaries of the ICL atlas (as this atlas is more accurately registered with the anatomy of the template brain). The mFG mask was taken directly from the UNC atlas. For the CAL mask, voxels that fell within the occipital lobe mask from the ICL atlas were included, and individual voxels falling outside of the calcarine cortex were manually removed. For the RGY and mSFG masks, only voxels that fell within the frontal lobe mask from the ICL atlas were included. For the IPL, voxels that fell within the parietal lobe mask from the ICL atlas were included. For the SMA, voxels labelled as CSF by the ICL atlas were removed.

Four brain regions were not included as independent brain regions in either atlas and were therefore hand-drawn. AI, PAG, PON and RVM masks were manually drawn in FSLeyes (FSL, Version 5.0.10). Adult masks of the AI (*Wiech et al., 2014*) and PAG (*Ezra et al., 2015*) were used to guide drawing of masks over the infant template brain. PON and RVM masks were drawn with reference to the Duvernoy Atlas (*Duvernoy, 2012*). The RVM mask fell within the region of the ventromedial nucleus of the solitary tract, while the PON mask consisted of the pontine nuclei and the basilar part of the pons (*Figure 1—figure supplement 1*).

Mean time series were calculated in EPI space by (i) registering masks of each region from the standard neonatal (40 week GA) template to EPI space (via structural scans) (*Figure 1—figure supplement 2A–C*), (ii) checking that each ROI mask fell within the field of view and was appropriately registered, (iii) using the function *fslmeants,* available in FSL, to generate a mean time series within each specific ROI (*Figure 1—figure supplement 2D*). Note, for the RVM (as this is a small ROI), we extracted a weighted time series using the mask weights in functional space. While masks were in structural space (*Figure 1—figure supplement 2B*), segmentations of individual participant's structural images (generated using a beta release of the developing Human Connectome Project (dHCP) pipeline) were used to remove voxels classified as CSF. Finally, voxels that fell within regions of signal dropout (i.e. where there was greater than 10 % signal loss in the functional image from maximum signal intensity) were also rejected. This was done by using a mask - automatically generated in FEAT - to identify and subtract voxels of large signal loss from the ROI masks.

All mean time series were demeaned and the three pre-stimulus data points were selected by identifying the volumes in which the noxious stimulus occurred and selecting the data from the three volumes immediately prior to each stimulus (see *Figure 3—figure supplement 2A*). The TR was 2.5 s so the pre-stimulus period had a duration of 7.5 s. As the stimuli could be applied at any time within a single volume the start point of the pre-stimulus period occurred between 7.5 and 10 s prior to the application of the stimulus. This resulted in 10 sets of 3 data points per mean time series per infant. Next, pairs of ROIs were taken from each infant's data (let these be ROI1 and ROI2). The overall pre-stimulus correlation between ROI1 and ROI2 was calculated by averaging the 10 pre-stimulus correlations (one per stimulus), which were the correlations between the three pre-stimulus points from ROI1 and the equivalent set of pre-stimulus points from ROI2. This method was repeated for all combinations of ROIs resulting in a connectivity matrix displaying all correlations per

infant (*Figure 3—figure supplement 2B*). Finally, the mean pre-stimulus functional connectivity was calculated per infant by averaging the below diagonal values of the connectivity matrix. The mean pre-stimulus connectivity was compared with the average post-stimulus percentage change in BOLD (calculated using the function *Featquery,* available in FSL) within a mask of all significantly active voxels ($z > 3.1$, $p < 0.05$) from the group analysis.

Regression analysis was carried out using MATLAB (Mathworks, version R2017a). Post-stimulus percentage change in BOLD in the group activity mask was input as the response variable into a linear regression model, which included the mean pre-stimulus functional connectivity within the network and infant's GA at study (in weeks and days) as the first and second explanatory variables respectively. The parameter estimates and p-values from the model are reported in the results. Finally, the r-value (Pearson's Correlation Coefficient) was calculated between the percentage change in BOLD and pre-stimulus functional connectivity (adjusted for age and obtained following the regression model fit).

## Acknowledgements

We thank the infants and their parents for taking part in this study. We also thank Olivia Faull, Jon Campbell and Michael Sanders for their thoughtful advice on the study design and analysis. This work was funded by the Wellcome Trust. Eugene Duff is a University of Oxford Excellence Fellow in Paediatric Neuroscience, supported by the SSNAP 'Support for the Sick Newborn and their Parents' Medical Research Fund.

## Additional information

### Funding

| Funder | Grant reference number | Author |
|---|---|---|
| Wellcome | 102176 | Fiona Moultrie |
| National Institute for Health Research | Clinical Doctoral Fellowship | Fiona Moultrie |
| Wellcome | Wellcome Centre for Integrative Neuroimaging, 203139/Z/16/Z | Irene Tracey |
| Wellcome | Senior Research Fellowship, 207457/Z/17/Z | Rebeccah Slater |

The funders had no role in study design, data collection and interpretation, or the decision to submit the work for publication.

### Author contributions

Sezgi Goksan, Conceptualization, Data curation, Formal analysis, Investigation, Methodology, Writing—original draft, Writing—review and editing; Luke Baxter, Formal analysis, Validation, Investigation, Methodology, Writing—review and editing; Fiona Moultrie, Validation, Investigation, Methodology, Writing—review and editing; Eugene Duff, Formal analysis, Supervision, Validation, Investigation, Methodology, Writing—review and editing; Gareth Hathway, Writing—original draft, Writing—review and editing; Caroline Hartley, Supervision, Validation, Methodology, Writing—original draft, Writing—review and editing; Irene Tracey, Conceptualization, Supervision, Investigation, Methodology, Writing—review and editing; Rebeccah Slater, Conceptualization, Supervision, Funding acquisition, Methodology, Writing—original draft, Writing—review and editing

### Author ORCIDs

Sezgi Goksan (iD) http://orcid.org/0000-0001-8836-0614
Luke Baxter (iD) http://orcid.org/0000-0001-9548-7162
Fiona Moultrie (iD) http://orcid.org/0000-0002-1431-791X
Eugene Duff (iD) http://orcid.org/0000-0001-8795-5472

Gareth Hathway [iD] http://orcid.org/0000-0003-4347-9667
Caroline Hartley [iD] http://orcid.org/0000-0002-7981-0836
Rebeccah Slater [iD] http://orcid.org/0000-0003-1595-4846

### Ethics

Human subjects: The National Research Ethics Service provided ethical approval: REC reference 12/SC/0447. Informed written parental consent, and consent to publish, was obtained. The study was carried out in accordance with the standards set by the Declaration of Helsinki and Good Clinical Practice guidelines.

### Decision letter and Author response

Decision letter https://doi.org/10.7554/eLife.37125.017
Author response https://doi.org/10.7554/eLife.37125.018

## Additional files

### Supplementary files
• Transparent reporting form
DOI: https://doi.org/10.7554/eLife.37125.014

### Data availability

The group brain activity data file is provided in the Source Data files. Raw data for individual infants is not provided as consent was not obtained for this data to be made publicly available.

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
