## [Decision Letter]

Thank you for submitting your article "The influence of the descending pain modulatory system on infant pain-related brain activity" for consideration by *eLife*. Your article has been reviewed by 3 peer reviewers, including Peggy Mason as the Reviewing Editor and Reviewer #1, and the evaluation has been overseen by Sabine Kastner as the Senior Editor.

The reviewers have discussed the reviews with one another and the Reviewing Editor has drafted this decision to help you prepare a revised submission.

The reviewers agree that these data are valuable, even precious, and that the manuscript is well written. The following aspects could be clarified or better justified.

– The control network. The reviewers discussed this "control" extensively. For the non-aficionados of fMRI, the inclusion of this network does not appear justified. What makes this collection of structures into a control? What is the functional connection between the structures or did you simply choose structures not involved in the pain matrix? The "control" network choices are further confusing given the group's previous work on hippocampus and pain (Ploghaus et al., 2001). In sum, this strategy needs to be better explained and justified.

– A different approach may be to replace the control network with a control time – a baseline period when activity is unrelated to the stimulus.

– There was skepticism regarding the resolution and accuracy of the brainstem activations. The authors may consider deleting this from the manuscript.

– The data are at odds with previous findings that facilitation predominates over inhibition in the neonate and that, consistent with this, the newborn human is exquisitely sensitive to nociceptor stimulation. Please address this discrepancy in the Results and Discussion.

– Descending modulation preferentially affects C-fiber inputs over A-deltas. A-delta inputs are often unaffected or facilitated by descending modulation. Again, this is at odds with the present results and the discrepancy should be discussed.

*Reviewer #1*:

This fMRI study in human infants suggests that functional connectivity within the pain matrix activates descending modulation resulting in less activation in the pain matrix. This study depends on highly processed fMRI data and a secondary analysis of this processed data from which connectivity is inferred. This reviewer is unable to assess the validity of these methods or the sufficiency or lack thereof of the sample size (13 subjects). Yet the ideas are intriguing, and the story is interesting and exciting.

*Reviewer #2*:

This fMRI study in human infants describes an inverse relationship between functional connectivity strength in networks associated with the descending pain modulatory system (DPMS) and nociceptor evoked brain activity. The authors suggest that in newborn infants the DPMS may contribute an inhibitory influence on nociceptor evoked brain activity.

I am not best placed to critically evaluate details of the fMRI design and methodology. However, I do have comments on the interpretation of the data.

1) Importantly the study includes a comparison between the relationship of DPMS network connectivity and a 'control' network to nociceptor evoked brain activity. However, it was unclear if there was a known functional relationship in the control network or whether these were just selected as brain regions not involved in the DPMS network.

2) The balance between facilitatory and inhibitory influences of DPMS on spinal nociceptive processing is dynamic and changes during the progression of chronic pain, in different behavioural and emotional states, and during development. There is a significant body of evidence to suggest that in neonatal rodents, facilitation predominates over inhibitory control and, consistent with this, the newborn human is exquisitely sensitive to nociceptor stimulation. However, this seems to be at odds with the data presented in this manuscript which imply that it is descending inhibition that is inversely related to the noxious-evoked brain activity i.e. in the newborn, the stronger the functional connectivity in components of the DPMS network the lower the evoked brain activity. The authors need to address this apparent anomaly.

3) For understandable ethical reasons, monitoring of brain activity is limited to responses to pin prick stimuli of cutaneous tissues, which will preferentially activate A-delta nociceptors. However, descending inhibitory control from the brain targets spinal neuronal responses to C-nociceptor stimulation, whereas responses to A-delta nociceptive input may be unaffected or even facilitated. The authors need to discuss the impact of this on the interpretation of their data.

*Reviewer #3*:

This manuscript details the results of an fMRI investigation examining noxious-stimulus evoked activation in response to pin-prick stimuli in a group of term born neonates. The authors examined pre-stimulus activation in brain regions mediating pain modulation. The activation in these brain regions was compared to the noxious-stimulus evoked activation and to activation in brain regions not involved in pain perception or its descending modulation.

The manuscript is part of new emerging literature examining the development of nociceptive pathways in infants using in vivo MRI. While the sample size is modest, the data obtained in the experiment are rare and were likely difficult to acquire.

This work is important to the field, the approach and results are novel and the manuscript is well written and straightforward. Some methodological issues affect the interpretation of results that I have detailed below. Further, the manuscript could be strengthened by adding to the Results and Discussion section to expand upon the interpretation of the findings.

Essential to the understanding of the results is the definition of the pre-stimulus period. While this information is included in the Materials and methods section and displayed in Figure 3—figure supplement 2, making mention of the timing and duration of the pre-stimulus period at the outset would be beneficial to the reader. Additionally, in the previous work by the group (Ploner et al., 2010), the pre-stimulus period was 3 secs before the noxious stimulus was administered. In the current work, what was the rationale for choosing the timing for the pre-stimulus period? Was there an indicator of when the noxious stimuli would be applied?

Related to the pre-stimulus event, was a pre-stimulus period modeled in the analysis for each stimulus including the first stimulus? Of interest would be to report on pre-stimulus activation throughout the course of the fMRI scanning run.

While it would not be possible to obtain pain ratings in the context of the current experiment with infants, the authors noted in their previous work (Goksan et al., 2015) that a foot withdrawal was often elicited in response to the stimulus. Did the authors record foot withdrawals during the course of the experiment?

What was the baseline (no pain condition) that the authors used for subtraction from the activation associated with the noxious stimuli? Figure 3—figure supplement 2A could be updated to highlight the baseline condition.

Imaging reliable activation in the brainstem in adults is challenging in the context of an fMRI experiment not involving noxious stimulation. As the authors note, obtaining reliable activation in the brainstem of an infant receiving noxious stimuli is not only difficult to due movement-related artifact but also by respiration. The authors note that they addressed motion in the current experiment by regressing out movement-related activation using ICA. The authors note that FIX was employed in the Materials and methods section. For FIX to be effective in identifying good and bad components, a training data set should be provided. Was this performed for the current analysis using a previous data set? Did the authors consider global signal reduction or "scrubbing" methods to address motion in the study?

Related to this issue of activation in the brainstem, the choice of blurring kernel of 4.5mm FWHM to perform spatial smoothing may be considered large in relation to the anatomical size of the brainstem nuclei. The size of the rostral ventral medulla (RVM) in adults is likely to be on the order of a few cubic millimetres. Did the authors consider the overall size of the RVM in the neonate during the preprocessing of the data? Was the activation reliable in this region across participants?

Of note is that no stimulus-evoked activation was seen in the anterior insula while instead activation in several frontal and temporal lobe regions. Could the authors add to the Results and Discussion section concerning the stimulus-evoked activation in the infant brain in relation to previously published experimental pain studies in the adult literature?

---

## [Author Response]

The reviewers have discussed the reviews with one another and the Reviewing Editor has drafted this decision to help you prepare a revised submissionThe reviewers agree that these data are valuable, even precious, and that the manuscript is well written. The following aspects could be clarified or better justified.– The control network. The reviewers discussed this "control" extensively. For the non-aficionados of fMRI, the inclusion of this network does not appear justified. What makes this collection of structures into a control? What is the functional connection between the structures or did you simply choose structures not involved in the pain matrix? The "control" network choices are further confusing given the group's previous work on hippocampus and pain (Ploghaus et al., 2001). In sum, this strategy needs to be better explained and justified.

Choosing a control network in fMRI studies is challenging. In this case, we chose a combination of brain regions that are not thought to be involved in descending pain modulation, and we ensured that they had a similar topographic distribution to the brain regions included in the DPMS network, consisting of cortical, subcortical and brainstem structures. Whilst we acknowledge that the hippocampus is involved in pain-related anxiety, our previous studies have not identified a role for the hippocampus in the DPMS (Ploghaus et al., 2001). The rationale for choosing this combination of regions was two-fold. First, to the best of our knowledge, this set of brain regions does not form a typical functional network, so there should be no relationship between the functional connectivity of this ‘network’ and the magnitude of the stimulus response, thus acting as a ‘negative’ control. Second, BOLD fMRI data is known to contain ‘global signals’ present across the entire brain that can originate from neurologically uninteresting sources, such as motion and respiration. Our data preprocessing included ICA denoising, which dramatically reduced the effects of both motion and respiration but cannot remove global effects. Inclusion of this ‘control network’ was intended to address the potential risk of an artefactual global signal driving the correlations we report – if a non-BOLD-related global signal were driving our results, this effect should be present in the Control Network – but this was not observed. We therefore consider that the inclusion of this network robustly demonstrates that general increased functional connectivity across the brain does not necessarily influence the noxious-evoked changes in BOLD activity. The rationale for this approach has been explained in more detail in the revised Results and Discussion section.

We do however appreciate that these brain regions do not form part of an established network. Therefore, we have included an additional control, the Default Mode Network, which is an established network that has been identified in adults and term infants (Doria et al., 2010; Raichle, 2015). This allows us to demonstrate the specificity of the relationship between the prestimulus functional connectivity of the DPMS and the noxious-evoked BOLD activity. The Default Mode network included the posterior cingulate cortex (PCC), medial prefrontal cortex (mPFC) and inferior parietal lobules (IPL).

These additional analyses have been added to the Results and Discussion section, where we now show that there is no evidence that the prestimulus functional connectivity of the Default Mode Network influences noxious-evoked BOLD activity. We would like to thank the Reviewers and Editors for raising this important point – we believe it has substantially improved the validity of our reported observations.

– A different approach may be to replace the control network with a control time – a baseline period when activity is unrelated to the stimulus.

We did not record resting state brain activity in this study so there is limited scope to investigate the relationship between the functional connectivity of the DPMS in an ‘independent’ time-period and the noxious-evoked BOLD activity. This is an important question and it forms a central direction for future research in this area, which we now discuss in the manuscript (subsection “Relationship between the pre-stimulus functional connectivity of the DPMS and noxious-evoked brain activity”).

– There was skepticism regarding the resolution and accuracy of the brainstem activations. The authors may consider deleting this from the manuscript.

We very carefully considered the inclusion of brainstem activations and the accuracy of localising them in the RVM and PAG and discussed this extensively with experienced adult brainstem imagers (Faull – included in the Acknowledgements), in addition to our authors who have published extensively on the adult brainstem and DPMS. To ensure that the results we report do not entirely rely upon the connectivity of these brainstem regions, we conducted the analysis both with and without their inclusion, confirming that the inclusion of brainstem data did not significantly influence the results.

To address the reviewers concerns further, we have now provided the anatomical RVM and PAG masks in functional space for each infant (Supplementary file 3), and an example time course for both regions from one infant can be seen in Figure 3—figure supplement 2A. While we acknowledge the inherent limitations of studying small brain regions, we have carefully considered the activity identified to ensure that it is localised within these anatomical structures. Data from four infants where brainstem data was not adequately localised were excluded from all analyses in the original manuscript.

We acknowledge that application of spatial smoothing, and variations in approaches of transforming small ROIs from standard to functional space, could potentially affect the results from these small regions. In order to address this, we have revised the Materials and methods section and now report data without spatial smoothing and using a weighted mean timecourse from the RVM ROI, with minimal impact on the results reported in the manuscript.

These limitations have now been specifically addressed in the Results and Discussion section.

– The data are at odds with previous findings that facilitation predominates over inhibition in the neonate and that, consistent with this, the newborn human is exquisitely sensitive to nociceptor stimulation. Please address this discrepancy in the Results and Discussion.

This is an extremely important point, which we did not adequately discuss in the original manuscript but is now explained in detail in our revision (Results and Discussion section). In brief, both human and non-human neonates are substantially more sensitive to noxious stimulation than adults, which has long been regarded as being the result of hyper-excitable spinal reflex networks. In the neonatal rodent, DPMS brain structures facilitate (rather than inhibit) spinally-mediated nocifensive behaviours. Similarly, in premature human infants, spinally-mediated reflex withdrawals are exaggerated. During human preterm development however, it has been shown that these reflex responses are refined, with shorter duration, lower amplitude and higher response thresholds (Cornelissen et al., 2013; Hartley et al., 2016). The data presented here in term infants suggest that by term gestation DPMS networks are present and appear to have an inhibitory function, evidenced by a correlative reduction in noxious-evoked brain activity, similar to that observed in the more mature adult brain. While it is tempting to draw comparisons between this study and those in neonatal rodents that have measured spinal responses to PAG activation, it must be acknowledged that our study has not measured activity in the spinal cord, and we do not know how increased functional connectivity in the DPMS impacts spinal activity in the term-aged infant. In order to address the neonatal rodent literature in our manuscript, we have added some caveats to the Discussion about assumptions regarding age equivalence between human infants and neonatal rat pups and assumptions regarding CNS maturation as a linear and uniform process. Different parts of the CNS clearly mature at different rates and the developmental trajectory may differ between rodents and man.

– Descending modulation preferentially affects C-fiber inputs over A-deltas. A-delta inputs are often unaffected or facilitated by descending modulation. Again, this is at odds with the present results and the discrepancy should be discussed.

As highlighted by the reviewer, in the adult rodent, PAG activation differentially modulates C- and A-delta sensory fibre dorsal horn input (Waters and Lumb, 2008), but how other supraspinal components of the DPMS respond to activity in subclasses of nociceptors is unknown, especially in man. Equally, the age at which descending fibres from the PAG-RVM axis and dorsal horn neurons form is unknown. This is now discussed in detail in the manuscript (Results and Discussion section).

Reviewer #2:This fMRI study in human infants describes an inverse relationship between functional connectivity strength in networks associated with the descending pain modulatory system (DPMS) and nociceptor evoked brain activity. The authors suggest that in newborn infants the DPMS may contribute an inhibitory influence on nociceptor evoked brain activity.I am not best placed to critically evaluate details of the fMRI design and methodology. However, I do have comments on the interpretation of the data.

Reviewer 2 has raised several important points regarding the interpretation of our data. These points were all highlighted by the Editors as key points to address. Each of these points have been revised in the manuscript and addressed directly in our response to the Editors (above). We thank the reviewer for this insightful review.

1) Importantly the study includes a comparison between the relationship of DPMS network connectivity and a 'control' network to nociceptor evoked brain activity. However, it was unclear if there was a known functional relationship in the control network or whether these were just selected as brain regions not involved in the DPMS network.

This has been discussed in detail in the revised manuscript and in the response to the Editor (see above).

2) The balance between facilitatory and inhibitory influences of DPMS on spinal nociceptive processing is dynamic and changes during the progression of chronic pain, in different behavioural and emotional states, and during development. There is a significant body of evidence to suggest that in neonatal rodents, facilitation predominates over inhibitory control and, consistent with this, the newborn human is exquisitely sensitive to nociceptor stimulation. However, this seems to be at odds with the data presented in this manuscript which imply that it is descending inhibition that is inversely related to the noxious-evoked brain activity i.e. in the newborn, the stronger the functional connectivity in components of the DPMS network the lower the evoked brain activity. The authors need to address this apparent anomaly.

This has been discussed in detail in the revised manuscript and in the response to the Editor (see above).

3) For understandable ethical reasons, monitoring of brain activity is limited to responses to pin prick stimuli of cutaneous tissues, which will preferentially activate A-delta nociceptors. However, descending inhibitory control from the brain targets spinal neuronal responses to C-nociceptor stimulation, whereas responses to A-delta nociceptive input may be unaffected or even facilitated. The authors need to discuss the impact of this on the interpretation of their data.This has been discussed in detail in the revised manuscript and in the response to the Editor (see above).Reviewer #3:This manuscript details the results of an fMRI investigation examining noxious-stimulus evoked activation in response to pin-prick stimuli in a group of term born neonates. The authors examined pre-stimulus activation in brain regions mediating pain modulation. The activation in these brain regions was compared to the noxious-stimulus evoked activation and to activation in brain regions not involved in pain perception or its descending modulation.The manuscript is part of new emerging literature examining the development of nociceptive pathways in infants using in vivo MRI. While the sample size is modest, the data obtained in the experiment are rare and were likely difficult to acquire.This work is important to the field, the approach and results are novel and the manuscript is well written and straightforward. Some methodological issues affect the interpretation of results that I have detailed below. Further, the manuscript could be strengthened by adding to the Results and Discussion section to expand upon the interpretation of the findings.

Many thanks for this insightful review. The Results and Discussion section has been substantially edited to provide a more in-depth interpretation of our results. In particular, we have discussed the value of including the Default Mode Network and Control Network in our analyses; discussed the difference between observations in the human and rodent literature; and expanded on the limitations in brainstem imaging. We have also provided more supporting data in Supplementary figures and Supplementary data files. All the text edits have been highlighted in the revised manuscript.

Essential to the understanding of the results is the definition of the pre-stimulus period. While this information is included in the Materials and methods section and displayed in Figure 3—figure supplement 2, making mention of the timing and duration of the pre-stimulus period at the outset would be beneficial to the reader. Additionally, in the previous work by the group (Ploner et al., 2010) the pre-stimulus period was 3 secs before the noxious stimulus was administered. In the current work, what was the rationale for choosing the timing for the prestimulus period? Was there an indicator of when the noxious stimuli would be applied?

The manuscript has been revised so that the key experimental details about the pre-stimulus period have now been included in the main body of the text (Introduction) and are further explained in the Materials and methods section. We have clarified that the pre-stimulus period included the three volumes prior to each stimulus. Given that the TR was 2.5 seconds, the duration of the pre-stimulus period was 7.5 seconds. The stimuli could be applied at any time within a volume, therefore the start of the pre-stimulus period ranged from 7.5 to 10 seconds prior to the application of the stimulus.

In this study, we chose a longer pre-stimulus period than previously used in the Ploner study to provide a more stable estimation of the time series correlations in the pre-stimulus period. Ten seconds represented the maximum number of time-points we could include without markedly encroaching on the recovery of the HRF, and allowed sufficient time for the stimulus evoked HRF to return to a baseline state (Arichi et al., 2012).

We have also clarified details concerning the application of stimuli in the Materials and methods section. Infants were unable to anticipate the stimulus as the experimenter received a visual cue 25 seconds after the stimulus and then waited for the infant to be still prior to applying the next stimulus. The application of stimuli was therefore not entirely regular or predictable.

Related to the pre-stimulus event, was a pre-stimulus period modeled in the analysis for each stimulus including the first stimulus? Of interest would be to report on pre-stimulus activation throughout the course of the fMRI scanning run.

An HRF model of the pre-stimulus period was not used in this study, but instead the mean time courses were extracted in the pre-stimulus period and the mean functional connectivity across the DPMS in the pre-stimulus period was calculated. We have extended our analysis to investigate whether the functional connectivity of this network was dependent on the stimulus number. We found that the pre-stimulus functional connectivity of the DPMS was stable and did not vary significantly. This result has been added to Figure 1—figure supplement 3 and included in the Results and Discussion section.

While it would not be possible to obtain pain ratings in the context of the current experiment with infants, the authors noted in their previous work (Goksan et al., 2015) that a foot withdrawal was often elicited in response to the stimulus. Did the authors record foot withdrawals during the course of the experiment?

During the experiment, we visually observed whether reflexes were evoked by the stimuli however we did not quantify these observations. Video footage of reflexes was not collected in this study, although in our current work we are now aiming to video infant reflexes throughout the scanning period. We have previously investigated the relationship between brain activity and reflex activity in infants using EEG and EMG, and we plan to investigate this further using fMRI. The importance of understanding this relationship is critical if we are to bridge our understanding from neonatal rat pup studies to human observations. This topic has been addressed in our revised Results and Discussion section.

What was the baseline (no pain condition) that the authors used for subtraction from the activation associated with the noxious stimuli? Figure 3—figure supplement 2A could be updated to highlight the baseline condition.

Our response to the stimuli was modelled with respect to the rest periods in between applications of the noxious stimuli (Figure 3—figure supplement 2). The inter-stimulus interval was a minimum of 25 seconds. This was not highlighted in the original figure and has now been added to the figure legend. The temporal mean that was used to calculate the percentage change in BOLD has been added to Figure 3—figure supplement 2D.

Imaging reliable activation in the brainstem in adults is challenging in the context of an fMRI experiment not involving noxious stimulation. As the authors note, obtaining reliable activation in the brainstem of an infant receiving noxious stimuli is not only difficult to due movement-related artifact but also by respiration. The authors note that they addressed motion in the current experiment by regressing out movement-related activation using ICA. The authors note that FIX was employed in the Methods. For FIX to be effective in identifying good and bad components, a training data set should be provided. Was this performed for the current analysis using a previous data set? Did the authors consider global signal reduction or "scrubbing" methods to address motion in the study?

We revised the Materials and methods section to include more comprehensive details regarding how we implemented FIX. In brief, the components were manually classified (Griffanti et al., 2017). FIX was then used to regress out both the noise ICA components and the 24 motion parameter time series. Thus, FIX was used to implement manual ICA clean-up, and did not need to be trained on any prior data. Also, using this approach manual FIX denoising can remove noise due to respiration and cardiac pulsatility, not just head motion. We did not use Scrubbing to remove time points that were influenced by motion artefact, as we found that implementing FIX provided a robust approach to reduce motion-related confounds. Similarly, while global signal regression (GSR) can be helpful in reducing motion artefacts, it also introduces confounds and can alter correlation structure and remove BOLD-related signal. We acknowledge that the data will contain global signals that cannot be removed by any of the preprocessing steps that we implemented, and therefore included the Control Network to address any global signal confounds (see response to Editor).

Related to this issue of activation in the brainstem, the choice of blurring kernel of 4.5mm FWHM to perform spatial smoothing may be considered large in relation to the anatomical size of the brainstem nuclei. The size of the rostral ventral medulla (RVM) in adults is likely to be on the order of a few cubic millimetres. Did the authors consider the overall size of the RVM in the neonate during the preprocessing of the data? Was the activation reliable in this region across participants?

We very carefully considered the inclusion of brainstem activations and the accuracy of localising them in the RVM and PAG and discussed this extensively with experienced adult brainstem imagers (Faull – included in the Acknowledgements), in addition to our authors with expertise in this field. To ensure that the results we report do not entirely rely upon the connectivity of these brainstem regions, we conducted the analysis both with and without their inclusion, confirming that the inclusion of brainstem data did not significantly influence the results.

To address the reviewers concerns further, we have now provided the anatomical RVM and PAG masks in functional space for each infant (Supplementary file 3) and an example time course for both regions from one infant can be seen in Figure 3—figure supplement 2A. While we acknowledge the inherent limitations of studying small brain regions, we have carefully considered the activity identified to ensure that it is localised within these anatomical structures. Data from four infants where brainstem data was not adequately localised were excluded from all analyses in the original manuscript.

We acknowledge that application of spatial smoothing, and variations in approaches of transforming small ROIs from standard to functional space, could potentially affect the results from these small regions. In order to address this, we have revised the Materials and methods section and report data without spatial smoothing and using a weighted mean timecourse from the RVM ROI, with minimal impact on the results reported in the manuscript.

Of note is that no stimulus-evoked activation was seen in the anterior insula while instead activation in several frontal and temporal lobe regions. Could the authors add to the Results and Discussion section concerning the stimulus-evoked activation in the infant brain in relation to previously published experimental pain studies in the adult literature?

We previously described that activity in the insular cortices was restricted to the posterior section (Goksan et al., 2015) and this observation is again reported in the present study, using our optimised data analysis protocols. As highlighted by the reviewer, highly localised clusters of activity were recorded in frontal and temporal lobe regions. In adults, attention and the threat of pain increase pain perception, and is associated with increased neural activity in the anterior insular cortex (Ploner et al., 2011, Wiech et al., 2010). We could postulate that infants do not evaluate or contextualise the nociceptive input in the same way as adults, which may contribute to the lack of activity within these regions. However, we cannot make statistical inferences about subthreshold activity.